# Retention and Transport of Nanoplastics with Different Surface Functionalities in a Sand Filtration System

**DOI:** 10.3390/nano14010032

**Published:** 2023-12-21

**Authors:** Hande Okutan, Gabriela Hul, Serge Stoll, Philippe Le Coustumer

**Affiliations:** 1Ecole Doctorale, Sciences et Technologies, Université de Bordeaux Montaigne, 33607 Pessac, France; hande-mahide.okutan@etu.u-bordeaux-montaigne.fr; 2Department F.-A. Forel for Environmental and Aquatic Sciences, University of Geneva, 1205 Geneva, Switzerland; gabriela.hul@unige.ch (G.H.); serge.stoll@unige.ch (S.S.); 3Department of Geological Engineering, Mugla Sitki Kocman University, Mugla 48000, Türkiye; 4Earth Sciences Department, Université de Bordeaux, 33615 Pessac, France; 5Bordeaux Imaging Center, Université de Bordeaux, CNRS-UAR3420–INSERM US4, 33000 Bordeaux, France

**Keywords:** drinking water, nanoplastics, transport, retention efficiency, sand filtration, saturated column

## Abstract

The efficiency of sand filtration was investigated in terms of the behavior of the nanoplastics (NPLs) with different surface functionalities. The initial condition concentrations of NPLs were varied, and their effects on retention and transport were investigated under a constant flow rate in saturated porous media. The behavior of NPLs in this porous system was discussed by considering Z- average size and zeta (ζ) potential measurements of each effluent. The retention efficiencies of NPLs were ranked as functionalized with amidine [A-PS]+ > with sulfate [S-PS]− > with surfactant-coated amidine [SDS-A-PS]−. The reversibility of the adsorption process was revealed by introducing surfactant into the sand filter system containing adsorbed [A-PS]+ at three different initial state concentration conditions. The deposition behavior on sand grain showed that positively charged NPLs were attached to the quartz surface, and negatively charged NPLs were attached to the edge of the clay minerals, which can be caused by electrical heterogeneities. The homoaggregates made of positively charged NPLs were more compact than those made of negatively charged NPLs and surfactant-coated NPLs. An anti-correlation was revealed, suggesting a connection between the fractal dimension (D_f_) of NPL aggregates and retention efficiencies. Increased D_f_ values are associated with decreased retention efficiencies.The findings underscore the crucial influence of NPL surface properties in terms of retention efficiency and reversible adsorption in the presence of surfactants in sand filtration systems.

## 1. Introduction

The ubiquity of plastics in common use constitutes an environmental pollutant attributed to the deficiency in implementing suitable recycling methodologies [1]. Nanoplastics (NPLs) and microplastics (MPLs) are emerging contaminants, and MPLs have been detected in freshwater sources such as surface water, groundwater, and wastewater [2,3,4]. The release of NPLs into the environment can occur either in their original nano-sized form during the manufacturing of cosmetic and cleaning products, paints, pharmaceuticals, and electronics or through the degradation and weathering of MPLs [5]. Therefore, NPLs can be present in natural water resources in their pristine, coated, or aged form, depending on their origin.

The size criterium of NPLs is plastic particles smaller than 1 µm in at least one of its dimensions [6]. Due to the dimensional characteristics of NPLs, differentiating them from other pollutants, such as MPLs, present in natural samples, the determination of their polymer type can be a challenging task. There is no standardized detection method for NPLs dispersed in natural matrices, leading to a lack of information about NPL concentrations in environmental samples [5]. In water treatment plants supplying drinking water, determining the NPL removal efficiency is vital because of the possible exposition of drinking water consumers to NPL particles.

The water treatment process is specific and depends on the drinking water treatment plant (DWTP) design. The main processes to remove dissolved and particle-based contaminants are coagulation/flocculation, sand filtration, activated carbon filtration, and disinfection [7,8,9]. Studies have been conducted to examine the effects of each process, individually or in combination, on the retention of NPLs.

Coagulants (polyaluminum chloride) were found to be more efficient, and their lower amounts were sufficient to successfully aggregate polystyrene (PS) NPLs of 90 nm [10]. The removal efficiency for NPLs was evaluated for the processes of coagulation/flocculation combined with sedimentation (CFS) and filtration. It was observed that the removal efficiency of NPLs (180 nm) with CFS treatment is insufficient [7]. The removal efficiency of sand filtration and activated carbon filtration processes for PS NPLs of 110 nm was found to be 88.1% [11]. The efficiency of sand filtration increased when the coagulation process was present. The removal capacity of the activated carbon filter was found to be more efficient than the one of the sand filter [11]. In contrast, the retention of sand filtration was found to be more prone to retain the NPLs than the activated carbon filter in the pilot-scale filtration unit. The difference may be caused by a biological active scum layer (schmutzdecke) in full-scale DWTPs [12]. It was revealed that by modifying granular activated carbon with zinc chloride (ZnCl_2_), the adsorption capacity of PS NPLs (100 nm) on granular activated carbon was significantly enhanced in the presence of lake water [13]. As for the disinfection process, it was stated that PS NPLs (100 nm) have resistance to chlorination, but they can be degradable with ozonation [9]. Ozonation with suitable doses at DWTPs resulted in negligible effects on the retention of NPLs during the subsequent filtration steps [12]. The studies show that the contribution of the filtration phases is undeniably high, which serves as an initial barrier, complementing other processes in the treatment plant [7,11,12]. Nevertheless, there is still a high probability of the presence of particles smaller than 1 µm in tap water, which cannot be retained by DWTP [9]. In order to prevent the presence of particles of such dimensions in potable water sources, it is crucial to examine the effects of different treatment processes with a specific emphasis on the sand filtration process that retains NPLs [14]. To manage and remediate the transport and fate of NPLs, which are in the environmental pollutant class, it is necessary to achieve a better understanding of the processes behind this event.

The retention and transport mechanism of NPLs in the sand filtration is expected to be controlled by (i) a collision mechanism: hydrodynamic forces, sedimentation, straining, interception, and Brownian diffusion; (ii) attachment mechanism: surface interactions potentials (van der Waals attraction and double layer repulsion); and (iii) detachment mechanism: change in flowrate and fluid chemistry [15,16]. The retention and transport behavior of NPLs in a saturated porous medium is usually investigated with column experiments, which can represent the sand filter of DWTPs. The increase in the ionic strength, particle size, and heterogeneity of porous media leads to an increase in the retention of negatively charged PS NPLs [17]. The initial concentration of positively charged PS NPLs played a crucial role in the adsorption capacity of sand grains [18]. The co-existence of NPLs with other contaminants and natural materials will affect the behavior of transport and deposition in subsurface environments or filter systems [19]. The pH and ionic strength conditions of the solution are also directly bonded to the behavior of materials co-existing with NPLs [20]. The clay minerals can be found as a part of porous media that causes an increase in the retention of negatively charged NPLs in quartz sand-type porous media [21,22]. The presence of surfactants in the water environment inhibits aggregation and affects the mobility of NPLs and MPLs by changing their charge [23,24,25].

The determination of fractal dimensions (D_f_) in this study serves to clarify the condition of particle aggregations. Fractal geometry has emerged as a valuable tool for elucidating the intricately irregular structures present in aggregated systems. It is commonly recognized that the application of the fractal concept offers a valuable understanding of how aggregation mechanisms affect the structure of aggregates [26]. The captured images from transmission electron microscopy (TEM) were used to analyze the fractal attributes of aggregates, and the D_f_ of nanoparticles was found in a range of 1.65 to 1.80 [27]. The study by Chakraborti et al. [28] involved a fractal dimension analysis employing image processing techniques for the time-dependent change in aggregation of monodisperse PS latex microspheres with the addition of coagulation. This analysis unveiled the D_f_ in a range of values between 1.94 and 1.48.

The sand filtration stage is a common process in treatment facilities. There are limited studies investigating the impact of sand filtration solely on the retention of NPLs with different surface functionalities. The assessment of the retention of NPLs with positive and negative surface functionalities during sand filtration, as well as the evaluation of the influence of surfactants—which constitute the primary raw materials of most cleaning agents—on both the retention and re-transportation of retained NPLs, highlight the unique contribution of this research. The assessment of the aggregation of the NPLs’ particles with fractal dimension calculations is another novelty of this study. Furthermore, obtaining comparable results for the behaviors of different NPLs within the same environment underscores the originality of this study.

Driving our research is the fundamental hypothesis that the efficiency of sand filtration in retaining NPLs is influenced by their surface functionalities, and this influence is further modulated by the presence of surfactant in the filtration system. The experimental conditions were integrated from the sand filtration system of the DWTP of Geneva (Switzerland) and were established under laboratory conditions with similar sand and flow rates conducted to explore the retention efficiency of the DWTP’s sand filtration with the presence of different surface functional groups, coating, and concentration of PS latex NPLs. The effect of surfactant on both adsorption and desorption processes was investigated. The emphasis of this study was particularly on (i) the effect of different surface properties of PS NPLs on their transport in porous media, (ii) the behavior of surfactant-coated PS NPLs during transport through the sand column, (iii) the reversibility of adsorption of positively charged PS NPLs with surfactant introduction into the sand filter, and (iv) the aggregation properties were examined with TEM images and assessed with fractal dimension calculations.

## 2. Materials and Methods

### 2.1. Materials

Surfactant-free PS latex NPLs with positive amidine functional groups -C(NH_2_)(NH_2_)+ on their surfaces were kindly provided by the Institute of the Analytical Sciences (ISA) affiliated with the National Centre for Scientific Research (Paris, France) and the University Claude Bernard (Lyon, France). The suspension contained 50 g/L of styrene transformed into PS with a conversion rate of 86%. The size of the positively charged amidine PS latex NPLs ([A-PS]+) was determined by dynamic light scattering (DLS), and TEM was equal to 177 ± 4 nm and 147 ± 17 nm, respectively. Surfactant-free sulfate latex NPLs with negative sulfate functional groups (${\mathrm{-SO}}_{4}^{2-}$) on their surfaces were purchased from Invitrogen (ThermoFisher Scientific, Basel, Switzerland). According to the manufacturer, they are characterized by a primary diameter of 0.1 µm, a density equal to 1.055 g/cm^3^ at 20 °C, and a specific surface area of 5.9 × 10^5^ cm^2^/g. The size measurements of the negatively charged sulfate PS latex NPLs ([S-PS]−) performed using DLS and TEM were comparable with the provider’s specification and equal to 114 ± 1 nm and 109 ± 10 nm, respectively. Both stock solutions were kept in a dark place at a temperature of 4 °C and were used to prepare solutions of lower concentrations. All dilutions were prepared in ultrapure water (Milli Q water, Millipore, Switzerland, with resistivity > 18 MΩ·cm and total organic carbon (TOC) < 2 ppb). A pH–titration curve was established for both types of NPLs to determine their characteristics in changing pH conditions. A 50 mg/L suspension of each NPL type was titrated with diluted 1 M HCl and 1 M NaOH solution (Titrisol, Merck, Schaffhausen, Switzerland) in a pH range of 3.0 ± 0.1 to 11.0 ± 0.1. pH changes were controlled using a Hach Lange HQ40d portable meter (Hach Lange, Geneva, Switzerland).

Sodium dodecyl sulfate (SDS; AppliChem, purity > 99%, Darmstadt, Germany) solutions were prepared by dissolving the surfactant in ultrapure water and used without further purification to prepare SDS-8-coated PS amidine latex NPLs ([SDS-A-PS]−). The latter was prepared by mixing PS amidine latex NPLs solution of a given concentration with an SDS solution of a given concentration. Critical SDS concentrations were determined experimentally to obtain a complete coating of NPLs surfaces and to avoid having free surfactants in the solution. Further information is given in Appendix A.

Fluoresceine sodium salt (Sigma Aldrich, Buchs, Switzerland) was used as an inert dye tracer to characterize the hydrodynamic parameters of porous media. A total of 1 g/L dye solution was prepared by dissolving a given amount of powder in ultrapure water. The stock solution and its dilutions were prepared right before use and protected from the direct influence of sunlight.

Quartz sand samples were provided by Carlo Bernasconi AG (Zürich, Switzerland), and they are composed of silica, i.e., 97–99%, along with other constituents in trace amounts, including Al_2_O_3_, K_2_O, Na_2_, TiO_2_, Fe_2_O, TiO, CaO, MgO, Fe_2_O_3_, and Na_2_O. The size and morphology characteristics of sand grains determined using Camsizer (Retsch, Haan, Germany) were kindly provided by the Industrial Services of Geneva (Appendix A). As reported before, sand grains carry a negative charge in a wide range of pH, i.e., from pH 1.0 ± 0.1 to pH 12.0 ± 0.1 [29]. The sand samples were not submitted to any chemical treatment or purification and were thoroughly rinsed with ultrapure water before use. No special cleaning procedure was applied to better reflect conditions prevailing in the industrial sand filters and the natural environment. The sand porosity was found at a value of 0.41 ± 0.1 with fluorescent dye tracer experiments and calculated with advection and dispersion equations. The details can be found in the Appendix A.

### 2.2. Methods

#### 2.2.1. Z-Average Hydrodynamic Diameter and Zeta (ζ) Potential Measurements

Z-average hydrodynamic diameter and zeta (ζ) potential of [A-PS]+, [S-PS]−, and [SDS-A-PS]− NPLs were determined with a Zetasizer Nano ZS (Malvern Instruments, Worcestershire, UK) using dynamic light scattering (DLS) and laser doppler velocimetry (LDV), respectively. All samples were measured after 1 min equilibration and at a temperature of 25 °C. The refraction index and absorbance for both types of PS NPLs were set to 1.59 and 0.01, respectively. Z-average hydrodynamic diameter and ζ potential calculations were based on five measurements, and each measurement consisted of ten runs. NPLs size was calculated using the Stokes–Einstein relationship [30], whereas ζ potential values were derived using the Smoluchowski approximation model [31].

#### 2.2.2. Turbidity Measurements

The turbidity of the effluents was measured using a Hach Turbidimeter model TU 5200 (Hach Lange, Geneva, Switzerland) in NTU units. A relation between NPL concentration and turbidity was established by plotting turbidity vs. a known concentration of NPLs. A set of samples with PS NPL concentrations in the range of 0.1 mg/L to 50 mg/L was prepared for each type of NPLs by diluting a given amount of stock solution in ultrapure water. The turbidity of each sample was verified after 15 min of stabilization at room temperature. In parallel, z-average hydrodynamic diameters and zeta potentials of NPLs were verified with the ZetaSizer Nano ZS (Malvern Panalytical, Chipping Norton, Australia).

#### 2.2.3. Scanning Electron Microscopy (SEM) and Transmission Electron Microscopy (TEM) Imaging

A JSM-7001FA (JEOL Ltd., Tokyo, Japan) scanning electron microscope (Department of Earth Sciences, University of Geneva, Geneva, Switzerland) and a Hitachi H7650 (Hitachi High-Tech Corp., Tokyao, Japan) transmission electron microscope (Bordeaux Imaging Center, Bordeaux, France) were used to obtain images of different types of NPLs. The Hitachi instrument was employed for High-Contrast Transmission Electron Micrography (HC-TEM) within a high vacuum, employing magnifications ranging from 40 k to 100 k and an accelerating voltage of 80 kilo-electron volts (keV). A detailed description of the sample’s preparation can be found in Appendix A.

#### 2.2.4. UV-Vis Spectrophotometry

UV-vis spectrophotometry was used to determine fluoresceine sodium salt concentrations. To find a relation between dye concentration and absorbance, a series of samples with dye concentration in a range of 0 to 20 mg/L was prepared by diluting a given amount of stock solution in ultrapure water, and the absorbance of each sample was measured at the wavelength of 460 nm. All measurements were performed in a 1 cm path-length quartz cuvette at room temperature by using a DR3900 spectrophotometer (Hach Lange, Geneva, Switzerland). A calibration curve was established according to the Beer–Lambert law and subsequently used to calculate dye concentrations during column experiments (Appendix A).

#### 2.2.5. Column Experiments

The transport of different types of NPLs in porous media was explored through wet-packed column experiments. As shown in Appendix A, the experimental setup consisted of the ultrapure water reservoir, silicon tubing, and acrylic column (36 mm inner diameter, 200 mm length of column) sealed with steel screens of mesh size of 250 µm and caoutchouc stoppers. An upward flow of ultrapure water was induced within the column using a peristaltic pump (Minipuls 3, Gilson, Mettmenstetten, Switzerland). The flow rate was adjusted to 20 ± 1 mL/min for each experiment. While filling the column with wet sand grains, the sides of the column were gently tapped with a stick to avoid air bubble entrapment and to help the stationary phase settle tightly and correctly. Prior to each experiment, the column was equilibrated to the steady state condition by flushing ultrapure water for at least 2 h. During the experiment, 0.3 mL of NPLs solution was injected into the column and followed by the background solution. The samples were collected manually every 60 s and 30 s after the NPLs injection. After the experiment, a 1 cm layer of sand was collected near the inlet, the middle, and the outlet of the column to characterize changes on the sand surface with SEM imaging. In selected columns, 0.3 mL of SDS solution was injected into the columns after the filtration stage to examine the influence of surfactants on the remobilization of retained NPLs. The effluent concentration of NPLs was determined with a turbidimeter, and the breakthrough curves were obtained by plotting the normalized concentration (*C*/*C*_0_) versus time. All experiments were performed at least in triplicate. Breakthrough curves were then used to calculate recovered mass according to the following equation [32]:(1)$$m_{R}=Q\int_{0}^{\infty }C\left(t\right)\partial t$$
where *m_R_* is recovered mass [M], *Q* [L^3^·T^−1^] is flow rate, and *C*(*t*) is the tracer concentration by the time [M·L^−3^]. The retention efficiency was obtained with the following equation:(2)$$R=\frac{\left(m_{i\;}-m_{R}\right)}{m_{i}}\times 100$$
where *R* represents retention efficiency, and *m_i_* represents the total injected mass [M] of NPLs.

#### 2.2.6. Aggregate Assessment with Fractal Dimension

The concept of fractal geometry was developed by B. Mandelbrot to understand the structural features and properties of various objects, such as colloidal aggregates [33]. Utilizing fractal geometry to analyze different characteristics of aggregates presents a valuable avenue for deepening our understanding of aggregation processes and refining modeling techniques. Fractal dimensions can be delineated in linear, planar, or volumetric terms, yielding corresponding one-dimensional, two-dimensional, or three-dimensional values, respectively [28]. The two-dimensional TEM images of aggregates obtained from the peak point effluents of selected experimental conditions were used and treated with the FRACLAC plugin of Image J FIJI 1.53t software [34,35]. The box-counting method was used, which consists of estimating the area-filling properties of the boundary at various scales by counting how many boxes, F, are required to cover the object in a grid of a given size, ε [36]. In other words, ε refers to the scale, which is the ratio of the box size to the image size. F is the number of pixels counted in per box [37]. This process is then repeated for multiple grids with increasing spacing. The slope of the log–log plot of ε (*x*-axis) versus F (*y*-axis) refers fractal dimension, -D_f_. The D_f_ value obtained expresses the two-dimensional fractal dimension value. To verify the methodology, the D_f_ of the Sierpinski Carpet was computed, yielding a projected D_f_ value of 1.88, while its theoretical value stands at 1.89 [38].

## 3. Results

### 3.1. Characterization of NPLs with Different Surface Functionalities

First, titration curves were established to characterize NPLs behavior in different pH conditions and are presented in Appendix A. Appendix A shows that [A-PS]+ NPLs are positively charged until reaching the point of zero charge, where the adsorbent’s net surface charge becomes neutral. This transition occurred at a pH value of 10.7 ± 0.1. Before charge neutralization, the mean z-average hydrodynamic diameter of [A-PS]+ NPLs varies around 177 ± 6 nm, and the mean ζ potential is equal to 43 ± 2 mV. After charge neutralization, the z-average hydrodynamic diameter increases to approximately 600 nm, and the ζ potential becomes negative and equal to −21 ± 1 mV. Appendix A demonstrates that [S-PS]− NPLs are stable and carry a negative charge for the whole range of pH. The mean z-average hydrodynamic diameter of [S-PS]− NPLs is equal to 116 ± 1 nm, and the mean ζ potential is equal to −43 ± 3 mV (Appendix A).

[A-PS]+ NPLs were coated with SDS for selected experiments. The required amount of SDS for coating was estimated with batch experiments. According to the results, 1000 mg/L, 500 mg/L, and 100 mg/L [A-PS]+ NPLs were completely coated in the presence of 500 mg/L, 400 mg/L, and 300 mg/L SDS concentrations, respectively.

When the NPLs are examined in terms of size according to TEM image analysis, the particles’ long and short axis ratio is close to 1. The short axis was preferred for calculations. [S-PS]− NPLs were observed as the smallest NPLs with a diameter of 109 ± 10 nm, followed by [A-PS]+ NPLs with a diameter of 147 ± 17 nm and [SDS-A-PS]− NPLs with a diameter of 168 ± 17 nm (Appendix A).

### 3.2. NPLs Transport and Retention Behavior in Sand Filter

The transport of NPLs was presented with normalized effluent concentration (C/C_0_) as a function of the effluent solution volume pore volume (PV) in Figure 1. The z-average hydrodynamic diameters and ζ potentials versus the effluent solution volume of each effluent of selected injection concentration were measured. The peak zone of C/C_0_ was shown at 0.5 and 1.2 PV for [A-PS]+ NPLs and at 0.5 and 1.4 PV for [S-PS]− and [SDS-A-PS]− NPLs. In the overall curve, this is the range where the most transportation of NPLs particles occurred (Figure 1).

The variation of injection concentration affected the retained NPLs amount in the porous media. When comparing the relationship between the retained mass percentages of [A-PS]+ NPLs and the injected concentrations, there was an uneven trend. In general, their retention percentages are close to each other. Here, the highest mass retention was observed for 500 mg/L injection with 79%, while the lowest retention was observed for 1000 mg/L injection with 71% (Table 1). The intermediate experiments with 150 mg/L and 200 mg/L were performed to know the correlation of retained mass values (Appendix A).

When the negatively charged particles were discussed, the effect of retention depending on the injected concentration was more noticeable than [A-PS]+ NPLs. The retention efficiency increased while increasing the injection concentration of both [S-PS]− and [SDS-A-PS]− NPLs. [SDS-A-PS]− NPLs were also observed to have higher transportation than [S-PS]− NPLs. The maximum retention was equal to 55% for [S-PS]− NPLs and 30% [SDS-A-PS]− NPLs for 1000 mg/L. At the minimum injection concentration of 100 mg/L, all the injected [SDS-A-PS]− NPLs were transported through the filtration column.

#### 3.2.1. Effects of Injection Concentration

As particles approaching the sand grain surface experience no repulsive forces, they come into physical contact with the sand grain surface and become immobilized. Favorable attachment conditions cause fast particle deposition. These particles typically do not detach as the strong, attractive forces exceed the hydrodynamic drag forces [39]. Unfavorable attachment conditions due to the presence of repulsive energy barriers lead to much slower particle deposition [40]. The breakthrough curves of [S-PS]− and [SDS-A-PS]− NPLs are shown in Figure 1b,c. The retention was observed in breakthrough curves significantly higher for [A-PS]+ than [S-PS]− and [SDS-A-PS]− NPLs (Figure 1).

For unfavorable attachment conditions, the increase in retention in porous media was caused by increasing the injection concentration of nanoparticles. It can be caused by the high collision probability, which occurs at high injection concentrations, and by the fact that already deposited NPLs provide additional retention sites [41,42].

In contrast, our results indicated that an elevated injection concentration leads to a decline in retention. This trend is commonly observed in micrometer-sized particles, as reported by Hou et al. [43] and under the consideration of the straining effect [44,45]. It was explained that the surface-active sites were filled more rapidly at higher concentrations, and progressive surface saturation caused a reduction in MPL retention in porous media [43]. In theory, when the ratio between the diameter of the particle and the median size of sand grains decreases, the mobility of particles decreases in saturated porous media. The effect of straining can be considered if this ratio is higher than 1.7 × 10^−3^ as the threshold [46]. In our investigation, the straining process may be deemed less efficient, primarily attributed to the size of the calculated threshold based on individual particles. However, it is essential to acknowledge that the formation of homo or heteroaggregates could potentially induce this type of straining effect.

#### 3.2.2. Effect of the NPLs Surface Properties

Z-average hydrodynamic diameter and ζ potential measured for NPLs present in the effluents are shown in Figure 2. For the [A-PS]+ NPLs, which is a favorable aggregation condition, there is an increase in ζ potentials while increasing the injection concentration (Figure 2b). The same phenomenon was observed for [S-PS]− and [SDS-A-PS]− NPLs for absolute values, which incorporate previous findings [47].

With the arrival of [A-PS]+ NPLs concentration, 0.75 PV, for 100 mg/L injection conditions, an increase in particle size was observed from 168 ± 3 nm to 350 ± 10 nm. For ζ potential measurements, a constant trend is observed at the natural level. It can be stated that when the electrostatic attraction between sand and NPLs decreases, transport rather than deposition of particles is expected. Charge neutralization can cause instability and the settling of the particles on the grain surface as homo- or heteroaggregated particles that can explain the observation of more retention according to the 1000 mg/L injection concentration (Figure 2a,b).

The impurities caused by the presence of clay minerals may influence the retention behavior of particles. Although the net charge of clay minerals is negative, the retention of NPLs increases due to the occurrence of positive or less negative deposition zones resulting from charge heterogeneities on clay minerals [21]. The clay minerals and their interactions with NPLs on the sand surface were imaged with SEM. The [A-PS]+ NPLs were generally deposited onto the quartz sand surface (Figure 3a). [S-PS]− NPLs were observed to be attached to the edge of clay minerals (Figure 3b). It should be noted that the used sand contains a low percentage of impurities, about 1–3%. Depending on whether the porous medium contains trace amounts of clay minerals, NPLs and clay particles can form heteroaggregates. In previous studies, it was observed that clay minerals adsorb at the MPLs surface and make charge differences in their surfaces in low ionic strength conditions [48]. The adsorption phenomena may lead to the measurement of neutrally charged values that we see at low injection concentrations of [A-PS]+ NPLs. Heteroaggregates can result in measuring a higher particle size.

For the injection concentration of 500 mg/L [A-PS]+ NPLs, the size of particles in injection suspension was 170 ± 2 nm. At the first arrival of particles, 0.62 PV, an increase in particle size (256 ± 16 nm) was observed, then sizes decreased to 200 ± 7 nm for the peak point and increased again. In terms of ζ potential, the charge, which was 53 ± 1 mV for the initial condition, reached a maximum of 26 ± 1 mV at the peak point and decreased up to 5 mV with a decrease in concentration. The trend in the ζ potential is reflected in the opposite way to the size values.

For the highest injection concentration, the initial injected size was 170 ± 2 nm, and at the peak point, it was measured at 184 ± 1 nm. Regarding the change in ζ potentials, 56 ± 2 mV was measured for the injection suspension, 30 ± 2 mV was measured at 0.63 PV, increasing to 46 ± 1 mV at 0.9 PV, and the lowest 24 ± 3 mV was measured at 1.2 PV. When the size and electrical charge are evaluated together, it can be said that a high electrical charge prevents aggregation and retention in the sand filter [49]. The lower stability observed for lower concentrations can be related to the lower magnitude of the particles’ ζ potentials [50].

Considering the measurements with negatively charged NPLs, the measured size of [S-PS]− NPLs in the injection suspensions is 115 ± 1 nm, 114 ± 1 nm, and 112 ± 1 nm for 100 mg/L, 500 mg/L, and 1000 mg/L, respectively (Figure 2c). For the three types of injection concentrations, an increase in size was observed with the first measurement of the concentration, a decrease in size at the peak, and an increase again with the decrease in the concentration. The size closest to the initial values was measured for the peak point with the injection concentration of 1000 mg/L. For ζ potential measurements, −35 ± 1 mV, −44 ± 1 mV, and −45 ± 1 mV, a strengthening trend with increasing concentration compared to the initial state. At 100 mg/L injection concentration, measurements started at −13 ± 1 mV and reached −23 ± 2 mV, PV 0.92, at the peak point (Figure 2d). The slight increases in z-average hydrodynamic diameter correspond with slight decreases in ζ potentials due to a reduction in electrostatic repulsion, as explained by DLVO theory [51].

Experiments with two intermediate concentrations, 150 mg/L and 200 mg/L, are available for [A-PS]+ and [S-PS]−. For [A-PS]+, the ζ potential measurements of these two values vary between 2 and 4 mV. The z-average hydrodynamic diameter measurements for 150 mg/L and 200 mg/L are close to the initial case and follow a similar behavior with 500 mg/L. For [S-PS]−, there is a similar trend between 100 mg/L, 150 mg/L, and 200 mg/L injection concentrations for ζ potentials. In size measurement, there is an increase according to the initial injection concentration conditions (Appendix A).

### 3.3. Effects of Surfactant Coating

In the case of [SDS-A-PS]− where negatively charged particles were formed by coating with SDS, initially, the sizes 172 ± 2 nm, 171 ± 2 nm, and 173 ± 2 nm were measured for 100 mg/L, 500 mg/L, and 1000 mg/L, respectively. As for the ζ potential measurements, the values measured for the injected suspension were −45 ± 1 mV, −45 ± 2 mV, and −44 ± 1 mV, respectively, from low to high injection concentrations (Figure 2e,f).

For 100 mg/L injection concentration, at 0.63 PV, the determined size was equal to 345 ± 2 nm, and at the peak point of 0.92 PV, it was equal to 308 ± 25 nm. After the peak point, the size increased. For 500 and 1000 mg/L injection concentrations, the same trend was observed, but the size increase was less and, at the peak, 0.92 PV, equal to 214 ± 6 d. nm for 500 mg/L and 220 ± 4 nm for 1000 mg/L. Unlike [S-PS]−, more differences were observed at 100 mg/L conditions compared to the other two conditions.

Positive values were measured in the first concentration measurements, then at peak concentrations, 0.92 PV, −7 ± 2 mV, −14 ± 1 mV, and −13 ± 1 mV for 100 mg/L, 500 mg/L, and 1000 mg/L were measured, respectively. The positive values can be explained by the presence of uncoated [A-PS] + NPLs. The surfactant coating on NPLs may undergo desorption when diluting the surfactant-NPLs suspension [52]. If the electrostatic force was greater than the adhesion force of the SDS molecules on the PS surface, this could be caused by the detachment of SDS molecules from the PS surface [24].

When the transport of NPLs injected with negative ζ potentials in two different ways is compared, it is observed that [S-PS]− NPLs are more strongly negatively charged than [SDS-A-PS]−, but in terms of transport, [SDS-A-PS]− was more transported. It could be concluded that the adhesive work of SDS and the decrease in surface tension increased the mobility of the NPLs.

### 3.4. Remobilization

The favorable attachment condition was examined in terms of desorption while injecting SDS solution into the sand filter after transport experiments at 100 mg/L, 500 mg/L, and 1000 mg/L initial concentrations of [A-PS]+ NPLs.

The recovery rates after both [A-PS]+ NPL and SDS solution injection were calculated. The maximum total recovered mass was measured for 100 mg/L injection conditions as 47%. The comparison between [A-PS]+ NPLs injection and SDS solution injection is shown in Table 2. The breakthrough curves of each condition are presented in Appendix A.

In the case of 100 and 500 mg/L concentrations, 16% and 10% were recovered after SDS solution injection. For the 1000 mg/L concentration, the recovered mass after the SDS solution injection is 7% of the retained mass were recovered.

After the injection of the SDS solution, the effluent samples initially consisted of positively charged particles, followed by negatively charged particles. As a result of the SDS solution injection, [A-PS] + NPLs became coated with SDS. This is followed by desorption due to the development of repulsive forces caused by negatively charged particles and sand (Figure 4a). Considering the z-average hydrodynamic diameter evolution, values close to the first case are observed for the peak zone for 500 mg/L and 1000 mg/L. For 100 mg/L, it can be assumed that aggregates or impurities are transported (Figure 4b). In addition, the low mass presence may have reduced the precision of the measurement. The increases in the standard deviations for all three concentrations examined indicate that larger impurities and possibly aggregates may have been transported at the end of the experiment due to the continuous flow.

### 3.5. Aggregate Properties

Samples were collected corresponding to the peak points at which experiments conducted with initial concentrations of 100 mg/L, 500 mg/L, and 1000 mg/L reached their maximum concentration. These selected samples were examined using TEM, and images describing the homoaggregates of NPLs were captured and utilized for the analysis of fractal dimensions.

In Figure 5, the relation of D_f_ values between injection concentration and NPL type was plotted, which were calculated from the slope of the log–log plot of ε vs. F (Appendix A). Given the studies indicating the fractal geometries exhibited by nanoparticle aggregates [27,28], fractal analyses were conducted to ascertain the presence of two-dimensional fractal geometries in the observed aggregates of NPL effluent samples examined via TEM.

In the case of [A-PS]+ NPLs, higher injection concentrations led to the formation of denser fractal aggregates. Conversely, for [S-PS]− and [SDS-A-PS]− NPLs, increased injection concentrations resulted in less compact fractal aggregates. This suggests that NPLs with varying ζ potentials exhibit contrasting aggregation behaviors in terms of compactness. For the lowest concentration, 100 mg/L, [S-PS]− and [SDS-A-PS]− NPLs showed denser aggregates than [A-PS]+ NPLs. When examining two negatively charged particles, the most notable distinction was evident in the scenario of 500 mg/L initial injection conditions. Here, it was observed that [S-PS]− aggregates exhibited higher compactness compared to [SDS-A-PS]− aggregates (Figure 5). The changes in D_f_ values observed during remobilization experiments involving the introduction of an SDS solution into the sand filter with adsorbed [A-PS]+ NPLs followed a nearly linear trend. As a result, in cases of remobilization, it can be deduced that alterations in the concentration of injected NPLs do not have a discernible impact on the outcome.

The phenomenon of homoaggregation is recognized to influence transport processes, leading to retention through straining effects and deposition on collector surfaces [53]. Considering the relationship between fractal dimension values and retained mass ratios, the impact of homoaggregation on retention was evaluated for the same type of NPLs (Table 3). For [S-PS]− and [SDS-A-PS]− NPLs, the variations in retained mass ratios (R_%_C_0_) and fractal dimension values (D_f_-C_0_) in relation to injection concentrations (C_0_) are outlined as follows: R_%1000_ > R_%500_ > R_%100_ and D_f100_ > D_f500_ > D_f1000_, respectively. When both of these assessments are considered collectively, it is observed greater retention efficiency in cases with less compact aggregates. In the case of [A-PS]+ NPLs, as with negatively charged particles, an increase in aggregate density led to a decrease in retention. However, in contrast to negatively charged particles, an increase in injection concentration resulted in higher compactness and lower retention in this context. The variation for [A-PS]+ NPLs can be shown as follows: R_%500_ > R_%100_ > R_%1000_ and D_f1000_ > D_f500_ > D_f100_. It is relevant to mention that the retained mass percentages exhibited a minimal difference between the 500 and 100 mg/L injection concentrations for [A-PS]+ NPLs. The increase in aggregate density, resulting in a decrease in aggregate size, subsequently led to a reduction in the retention percentage.

### 3.6. Summary of Mechanism

The mechanisms of NPLs retention in sand filters are schematically illustrated in Figure 6. In a stable state, three different types of NPLs were introduced into quartz sand-type saturated porous media representing the sand filtration phase. Due to the attractive forces between the quartz sand and [A-PS]+ NPLs resulting from their opposite charges, the retention of [A-PS]+ NPLs was higher compared to [S-PS]− and [SDS-A-PS]− NPLs, which have repulsive interactions. Surfactant-coated negatively charged [SDS-A-PS]− NPLs were retained less than function group-dependent negatively charged [S-PS]− NPLs. Due to charge heterogeneities, negatively charged particles are attached to the corners of clay minerals. Positively charged particles, on the other hand, are attached to quartz minerals. With the injection of surfactant into the sand filter containing adsorbed [A-PS]+ NPLs, a charge inversion occurred, causing the positively charged particles present in the media to undergo transport in a negatively charged state.

## 4. Conclusions

Transport experiments were performed to investigate the behavior of NPLs in a laboratory-based sand filtration column with similar sand and flow rate properties of the DWTPs of Geneva (Switzerland). The impact of surface functionality, as well as the presence of surfactant in both coating and solution forms, on adsorption and desorption in porous media was examined through measurements of z-average size and ζ potential. The breakthrough curves were established with turbidity measurements for the calculation of retention and transport efficiencies. The aggregation definition and comparisons were made with fractal dimension analysis from TEM imaging.

Our research highlights that the surface charges of NPLs, resulting from various surface functionalities or coatings with surfactant molecules, yield distinct and crucial impacts on the retention behavior within porous media. In cases where the NPLs surface charge is identical to the porous media (here, quartz sand grains) or the presence of a surfactant surface coating, retention is limited if the comparison is made with oppositely charged NPLs. The average retention efficiency of [A-PS]+, [S-PS]−, and [SDS-A-PS]− NPLs were 75%, 40%, and 15%, respectively, across all injection concentrations. This means that electrostatic interaction forces play a major role. Different surface functionalities of NPLs result in varying degrees of retention within saturated porous media during sand filtration. Despite both [S-PS]− and [SDS-A-PS]− NPLs initially bearing a negative charge, those particles coated with an anionic surfactant, [SDS-A-PS]−, exhibited lower retention in the sand filter. The compactness of aggregates has effects that correspond to the retention efficiencies. The density of aggregates exhibits effects that align with retention efficiencies. The results interpretation of the homoaggregation of the NPLs in the effluents showed that [A-PS] + NPLs are forming more compact aggregates than the [S-PS]− and [SDS-A-PS]− NPLs. For NPLs with similar characteristics, an increase in fractal dimension values, which signifies greater aggregate compactness, was observed to lead to a decrease in retention efficiencies. This increased compactness can be attributed to the smaller aggregate sizes, providing an explanation for the decrease in retention efficiencies.

The ζ potential demonstrated an obvious rise in positively charged particles as particle concentrations increased. However, in contrast, the effect of concentration variations on the negatively charged particles was less noticeable compared to positively charged particles. This may have also increased the retention of positively charged particles. It has been shown that particles retained in the sand filter can be released by the introduction of surfactant in the sand filter with retained NPLs. This remobilization was observed as a maximum transport of 35% from the retained [A-PS] + NPLs compared to the initial situation.

The sand filtration phase is essential for eliminating suspended particles and constitutes an integral component of DWTPs. It serves as a protective barrier, preventing particles from entering more delicate treatment processes like disinfection, where they might block or diminish the effectiveness of these procedures. This phase is not only cost-effective but also aligns with sustainable water treatment practices, as it operates through a natural process without reliance on chemicals, making it an environmentally friendly choice. Investigating the retention effects of sand filters on NPLs holds significant importance, both for refining sand filter performance and for guiding research toward enhancing the retention of unretained components in other stages of DWTPs. Additionally, it has come to light that the remobilization rate of retained NPLs, which may be transported into the subsequent phase with the introduction of pollutants such as surfactants, is a noteworthy consideration. Exploring whether other contaminants induce similar behavior warrants further inquiry. From a future perspective, developing a sand filter media with physical and chemical conditions closer to the natural environment presents an intriguing avenue of exploration.

## Figures and Tables

**Figure 1 nanomaterials-14-00032-f001:**
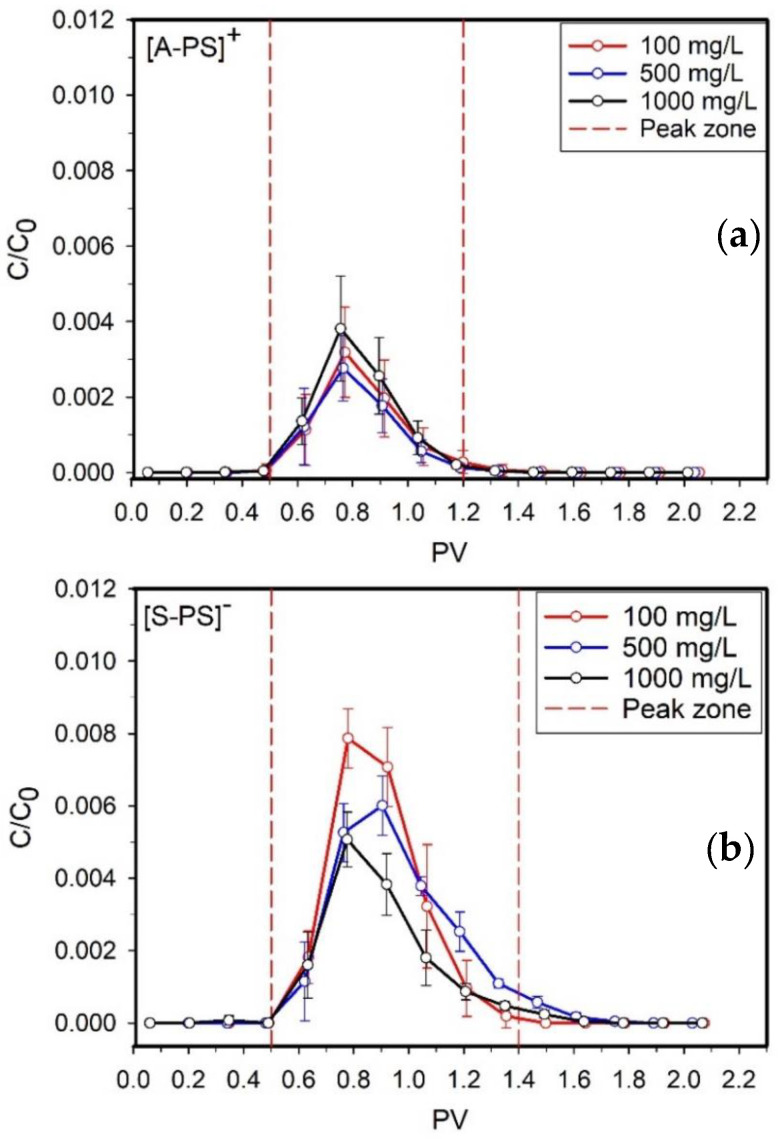
Breakthrough curves for 100 mg/L, 500 mg/L, and 1000 mg/L injection concentration (C_0_) are shown for [A-PS]+ (**a**), [S-PS]− (**b**), and [SDS-A-PS]− (**c**) NPLs.

**Figure 2 nanomaterials-14-00032-f002:**
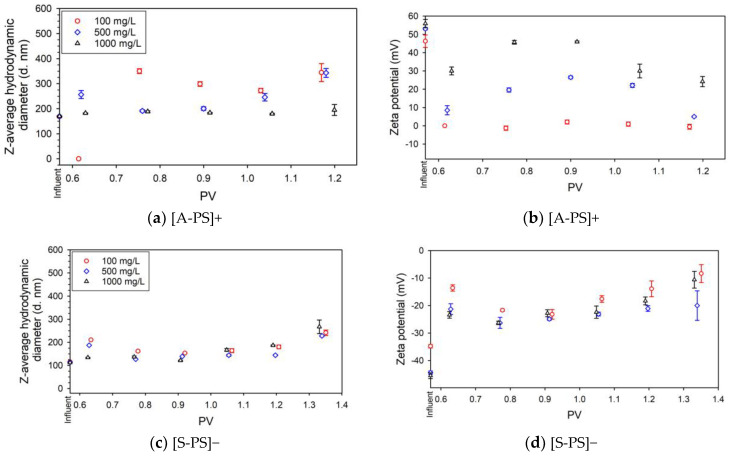
Collective representation of hydrodynamic diameter and zeta (ζ) potential vs. PV graphs of experiments performed with [A-PS]+ (**a**,**b**), [S-PS]− (**c**,**d**), and [SDS-A-PS]− (**e**,**f**).

**Figure 3 nanomaterials-14-00032-f003:**
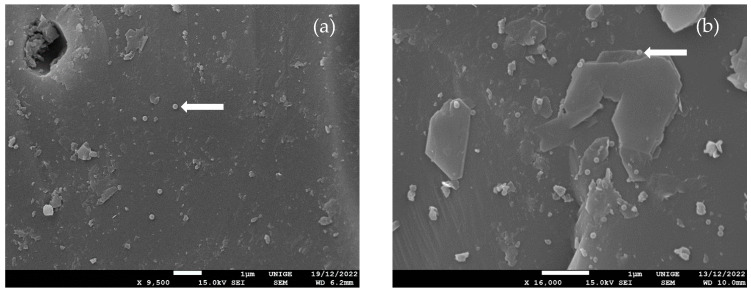
SEM micrography from the sand grain collected after the transport experiment. (**a**) Attachment of [A-PS]+ NPLs onto the quartz surface and (**b**) attachment of [S-PS]− NPLs onto clay minerals (white arrows).

**Figure 4 nanomaterials-14-00032-f004:**
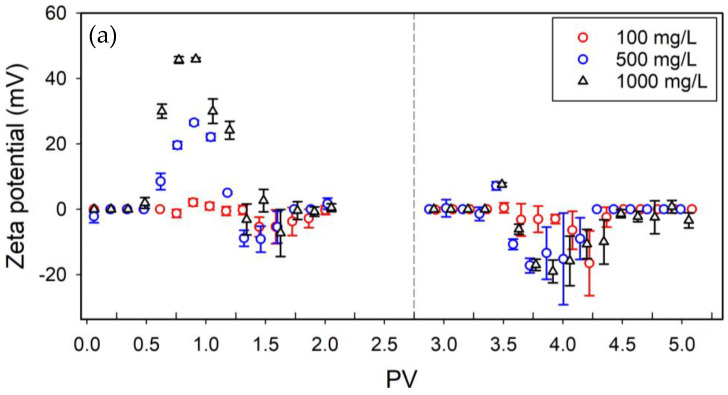
ζ potential measurements (**a**) and z-average hydrodynamic diameter (**b**) of effluents. The dashed line represents the SDS solution injection.

**Figure 5 nanomaterials-14-00032-f005:**
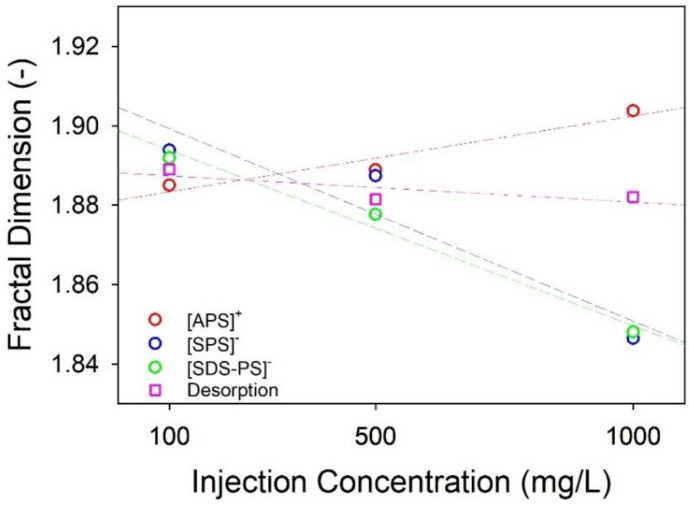
Fractal analyses results for each concentration, type of NPL, and desorbed sample. Three images of aggregates were used for each condition.

**Figure 6 nanomaterials-14-00032-f006:**
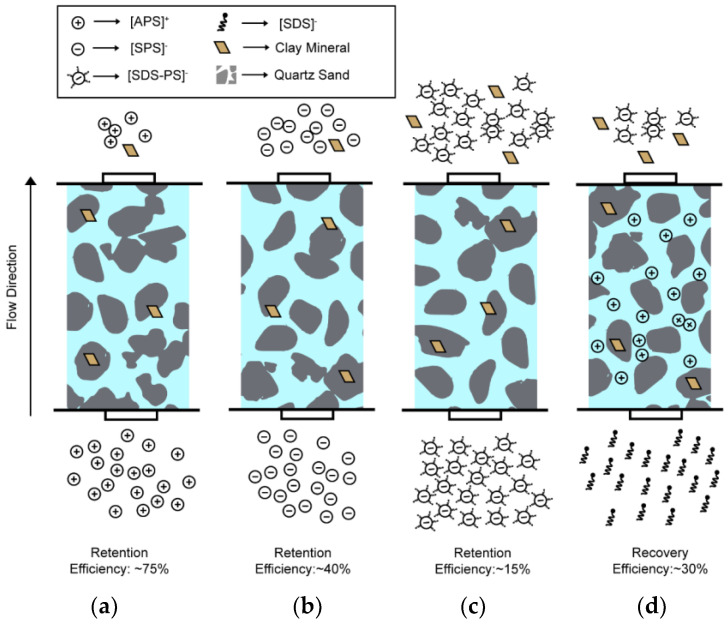
The schematic overview depicts the transport and retention process of NPLs within a sand filtration column which is quartz sand and characterized as a saturated porous medium. Panels (**a**), (**b**), and (**c**) illustrate the retention efficiencies for [A-PS]+, [S-PS]−, and [SDS-A-PS]− NPLs, respectively. Panel (**d**) presents the re-mobilization of adsorbed [A-PS]+ upon the introduction of an SDS solution.

**Table 1 nanomaterials-14-00032-t001:** Summary of injection conditions, maximum effluent concentrations (MEC), and retained mass values.

Type	Injection Concentration (mg/L)	MEC (mg/L)	Retention Efficiency (%)
[A-PS]+	100	0.4 ± 0.2	76 ± 9
	500	1.4 ± 0.4	79 ± 6
	1000	3.8 ± 1.2	71 ± 9
[S-PS]−	100	0.8 ± 0.1	32 ± 4
	500	3.1 ± 0.2	34 ± 2
	1000	5.1 ± 0.6	55 ± 3
[SDS-A-PS]−	1000	1.0 ± 0.1	3 ± 11
	100	4.4 ± 0.4	9 ± 10
	500	7.4 ± 1.0	30 ± 13

**Table 2 nanomaterials-14-00032-t002:** Total recovered mass from the experiments and the percentage of this total mass recovered after [A-PS]+ injection and SDS injection are shown. Recovered mass 1 represents the recovered mass percentage after NPL injections. Recovered mass 2 represents the recovered mass percentage after SDS solution injection, which is expected to cause remobilization of the sorbed NPLs. The total recovered mass percentage shows the recovered percentage according to initial conditions after both injections.

Injected NPL Concentration (mg/L)	Recovered Mass 1 * (%)	Recovered Mass 2 ** (%)	Total Recovered Mass (%)
100	31 ± 6	16 ± 4	47 ± 8
500	20 ± 6	10 ± 2	30 ± 5
1000	34 ± 8	7 ± 2	42 ± 6

* After [A-PS]+ NPL injection, ** After SDS solution Injection.

**Table 3 nanomaterials-14-00032-t003:** The overall presentation of retention efficiency and fractal dimension values.

Type	Injection Concentrations	Fractal Dimension, D_f_	Retention Efficiency, %
[A-PS]+	100	1.885 ± 0.017	76 ± 9
[A-PS]+	500	1.889 ± 0.019	79 ± 6
[A-PS]+	1000	1.904 ± 0.003	71 ± 9
[S-PS]−	100	1.894 ± 0.026	32 ± 4
[S-PS]−	500	1.887 ± 0.012	34 ± 2
[S-PS]−	1000	1.846 ± 0.005	55 ± 3
[SDS-A-PS]−	100	1.892 ± 0.005	3 ± 11
[SDS-A-PS]−	500	1.878 ± 0.015	9 ± 10
[SDS-A-PS]−	1000	1.848 ± 0.017	30 ± 13
Desorption	100	1.889 ± 0.022	52 ± 8
Desorption	500	1.881 ± 0.014	70 ± 5
Desorption	1000	1.882 ± 0.015	58 ± 6

## Data Availability

Data will be made available upon request.

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
