# Peer review of "Retention and Transport of Nanoplastics with Different Surface Functionalities in a Sand Filtration System"

_nanomaterials, 2023, doi:10.3390/nano14010032_

Round 1
Reviewer 1 Report
Comments and Suggestions for Authors
The manuscript entitled “Retention and transport of nanoplastics with different surface functionalities (amidine, sulfate, and surfactant coated amidine) in a sand filtration system” by Hande Okutan et al. deals with the investigation on retention of PS latex NPLs with positive and negative surface functionalities and different concentrations of PS latex NPLs under laboratory conditions, as well as fractal dimension calculations. The experimental conditions were adopted from the sand filtration system of the Geneva DWTP, using similar sand and flow rates. The concept and methodology are innovative, having a potential practical application for eliminating suspended particles of DWTPs. As methods, the authors used Z- average size and zeta potential, SEM, TEM, turbidity measurements, UV-vis spectrophotometry, and column experiments.
However, I would like to address some corrections/suggestions for improving the quality of the manuscript.
1) The Introduction should be improved with the definition of NPLs and their sources.
2) What were the outcomes of the experiments from the sand filtration system of the
DWTP of Geneva (Switzerland)? The NPLs were reduced as concentration?
3) Line 128: “positive amidine functional groups (-NH-NH2+)” - Please verify the name of this class (amidine or hydrazine?). Amidines are organic compounds with the functional group RC(NR)NR2, where the R groups can be the same or different. Hydrazine is an inorganic compound with the chemical formula N2H4.
4) Line 134: What was the concentration of surfactant-free sulfate latex NPLs with negative sulfate functional group?
5) Line 151: What were the concentrations of SDS solutions? They appear at Line 270.
6) From Material Section are missing the matrix characteristics used. What were the concentrations of positive and negative NPLs in water for the zetasizer measurements? It was interested if the authors would be tested the unmodified PS latex NPs, as comparison. Could the authors provide information on the parameters which were necessary for configuring the z-average hydrodynamic diameters and zeta potentials measurements (i.e. refraction index, absorbance, viscosity)?
Minor corrections:
Line 56: ”PS NPLs of 110 nm was found to be 88.1 %” should be supported by a reference.
Line 63, Line 166: - use subscripts
Line 68: “The studies show that the contribution of the filtration phases to the other phases is undeniably high” - it is unclear, please reformulate.
Line 70: “To prevent particles of this size presence in consuming water sources, there is a need to focus on investigating the effect of different treatment processes, especially the sand filtration process, on the retention of NPLs” - please rephrase it for a better understanding. For example, “To prevent the presence of particles of this size in consuming water sources, it is necessary to investigate the effects of different treatment processes, with special emphasis on the sand filtration process, to assess its efficiency in retaining NPLs”.
Line 137: check the numeric values for superscript.
Line 142: explicitly “R”
Line 148-150: is repeated at Line 178-180.
I recommend the authors to use the abbreviation form of words after its first description.
The equations should be numbered in a consecutive mode.
Please define the “point of zero charge”.
Line 316 - The ”.” is missing.
Author Response
Please find the responses in the attachment.

Reviewer 2 Report
Comments and Suggestions for Authors
A brief summary:
In reviewed paper the authors focused on the problem of the nano and microplastic particles present in water. The problem is important due to huge increase of use of plastic products in all branches of industry, which finally increase of concentration of plastic particles in environment. Due to the lack of intensive investigations we still not sure what impact on human health those impurities may affect so the increased investigations on the techniques of their separation and removal form water are absolute needed. In the water environment which globally can be described as a colloidal system, the surface characteristics of particles and interactions between them governing their behavior. The authors decided to focus on the influence of the surface characteristic of the plastic particles on the filtration process realized in sand column. I think the choice of the subject is correct and well recognition of the physic- chemical interaction in the system a kay part of the investigations. They choose the conditions of experiments integrated from the sand filtration system of the DWTP of Geneva (Switzerland). It is very proper way of doing experiments. The authors investigated the effect
of different surface properties of particles on their transport in porous media, the effect of surfactant-coating of particles on the filtration process and the aggregation properties of the particles assessed with fractal dimension calculations. All these goals are properly chosen but I did not find any scientific hypothesis stated in introduction. In my opinion the scientific paper should have one, and during investigation this hypothesis is defended or overthrew.
The authors very well described all materials and methods which were used during investigations. All used techniques are up to date, and I did not find any misleading’s. In the conclusions authors did not put any probe of explanation of the results. In this part of the paper authors collect only the observations but in my opinion they should, even in hypothetical way propose some interpretation. For each who has contact with colloidal systems obtained results are obvious, yes surface charge of particles yields distinct and crucial impacts on the retention behavior within porous media, small aggregates are deposit in porous media in less efficiency and ζ potential and presence of surfactant will affect on filtration in porous media. Globally speaking the article is interesting but more like the report form experiments than scientific dissertation…
Specific comments:
1) The authors should put in the introduction the scientific hypothesis of this work and in the conclusion in the light of obtained results stated if it is defended or overthrow and try to explain why …
2) The primary particles in colloidal systems may create the aggregates which has complex structure called fractal like aggregates. Aggregates are not a fractal in mathematical sense, due to its finite construction from single primary particles we can only use the fractal geometry concept to describe their morphology, mainly speaking their compactness. Aggregates are 3D structures, and pictures obtained using TEM microscopy are only 2D representation of their morphology. Verifying the methodology of evaluation Df called fractal dimension using Sierpiński Carpet proof only that this methodology work for 2D structures. May be to verifying the algorithm used bi ImageJ software would it be mor proper to use projections of 3D fractals on 2D plane?
Authors should put in the text information about it.
3) On figure 4a, authors should put the legend even if it is the same as in fig 4b
Author Response

(The authors gave the same response as above.)

Reviewer 3 Report
Comments and Suggestions for Authors
The efficiency of sand filtration was investigated in terms of the behavior of the nanoplastics with different surface functionalities. The authors explored the retention efficiency of the DWTP’s sand filtration with the presence of different surface functional groups, coating, and concentration of polystyrene latex nanoplastics.
1. There are same formatting errors, such as
“Accord ing to the manufacturer, they are characterized by a primary diameter of 0.1 μm, a density equal to 1.055 g cm-3 at 20°C, and a specific surface area of 5.9 · 105 cm2/g.”
“Quartz sand samples were provided by Carlo Bernasconi AG (Zürich, Switzerland) and they are composed of silica, i.e., 97 – 99 %, along with other constituents in trace amounts including Al2O3, K2O, Na2, TiO2, Fe2O, TiO, CaO, MgO, Fe2O3 and Na2O.”
……………
2. The writing of equation (2) is not correct.
3. The authors mentioned that the efficiency of sand filtration was investigated in terms of the behavior of the nanoplastics with different surface functionalities. In addition, the retention efficiencies of NPLs were ranked as functionalized with amidine [A-PS]+ > with sulfate [S-PS]- > with surfactant coated amidine [SDS-A-PS]-. If possible, please supplement the results that can prove the existence of these surface functionalities.
4. The authors should highlight this work.
5. The authors should provide more analyses on the mechanisms of nanoplastics retention in sand filters.
Comments on the Quality of English LanguageMinor editing of English language required.
Author Response

(The authors gave the same response as above.)
